# Analysis of Three Species of *Cassipourea* Traditionally Used for Hypermelanosis in Selected Provinces in South Africa

**DOI:** 10.3390/ijms25010237

**Published:** 2023-12-23

**Authors:** Nomakhosi Mpofana, Masande Yalo, Nceba Gqaleni, Ncoza Cordelia Dlova, Ahmed A. Hussein

**Affiliations:** 1Nelson R Mandela School of Medicine, Department of Dermatology, University of KwaZulu-Natal, Durban 4000, South Africa; nomakhosim@dut.ac.za (N.M.); dlovan@ukzn.ac.za (N.C.D.); 2Department of Somatology, Durban University of Technology, Durban 4000, South Africa; 3Department of Chemistry, Cape Peninsula University of Technology, Cape Town 8000, South Africa; yalom@cput.ac.za; 4Discipline of Traditional Medicine, University of KwaZulu-Natal, Durban 4000, South Africa; gqalenin@ukzn.ac.za; 5Faculty of Health Sciences, Durban University of Technology, Durban 4000, South Africa

**Keywords:** medicinal plants, tyrosinase inhibition, melanin inhibition, hypermelanosis, cosmetics, skin lightening

## Abstract

There is a growing demand and use of herbal cosmetics for skin purposes due to their perceived safety when applied to the skin. Three *Cassipourea* species commonly known as “ummemezi” are used interchangeably by women in rural areas of Eastern Cape and KwaZulu-Natal provinces to treat hypermelanosis as well as sun protection. We conducted a phytochemical comparison of three *Cassipourea* species; *Cassipourea flanaganii* (Schinz) Alston, *Cassipourea gummiflua* Tul. verticillata (N.E.Br.) J. Lewis and *Cassipourea malosana* (Baker) Alston by Liquid Chromatography–Mass Spectrometry (LC-MS/MS) analysis in negative mode. The results obtained from the LC-MS/MS yielded a total number of twenty-four compounds of different chemical classes, including fatty acids, steroids, di- and tri-terpenoids, flavonoids, phenolic acids, and eighteen among them were tentatively identified. The LC-MS /MS analysis showed that the three studied *Cassipourea* extracts contain compounds that have anti-tyrosinase activity and consequently. The presence of these compounds, either in synergy or individually, can be attributed to the anti-tyrosinase effect. Although the traditional names of the species are used interchangeably, they are different, however, they possess similar skin-lightening properties. Despite the recent popularity of modern cosmetic products, plants continue to play an important role in the local cosmetics industry in South Africa’s Eastern Cape and KwaZulu-Natal community provinces.

## 1. Introduction

As global acceptance grows, the use of medicinal plants to treat a variety of human diseases is no longer considered an antiquated practice. This may be attributed to their assumed safety, efficacy, affordability, and absence of side effects. However, most medicinal plants used in conventional medicine have not had their toxicological profiles evaluated [1,2,3]. According to The World Health Organisation (WHO), about 80% of the world’s population living in impoverished countries depend on the use of medicinal plants as a source of primary health care for treating and preventing various diseases and disorders [4]. It is estimated that 60–90% of the African population uses natural medicinal plants to treat ailments, owing to the availability and affordability of these medicines in comparison to popular or conventional biomedicines [5,6,7]. Due to its geographic location, Africa is a very hot country with daytime ambient temperatures that often exceed 35 °C [8]. In South Africa, levels of ambient solar, Ultra Violet Radiation(UVR) throughout most of the year are high with the Ultra Violet Index (UVI) being frequently extreme (11+ or > 6400 Jm^−2^/day) [8]. Thus, some plants are used for photoprotection as well as complexion enhancers [9,10,11,12].

Likewise, in South Africa, medicinal plants or traditional remedies are culturally and economically important resources for a large proportion of South Africa’s population [11,13,14]. South Africa accounts for 9% of the higher plants worldwide due to its rich cultural biodiversity, with over 30,000 plant species currently used by over 200,000 traditional healers in the prevention, treatment, and cure, of many diseases and skin disorders [1,15,16,17]. About 80% of the entire population in South Africa, particularly those who dwell in rural areas, adopt the use of medicinal plants in various forms as medicines for the maintenance of their health [2]. Some plants are used cosmetically to maintain healthy skin, such as improving skin complexion, skin lightening, depigmentation, Ultra Violet (UV) protection, sunburn treatment, treating various skin conditions such as breakouts, spot removal, and thus healing, restoring and skin moisturizing [18,19,20]. Previous surveys have revealed the bark is the most frequently used part of the plant [10,14,18].

## 2. Results and Discussion

Three *Cassipourea* species commonly known as “ummemezi obomvu” or “umqonga” are used interchangeably by women in rural areas of Eastern Cape and KwaZulu-Natal provinces to treat hypermelanosis as well as sun protection.

The species were identified as follows:Specimen 2: NH0151949-0, *Cassipourea malosana* (Baker) Alston (Figure 1).Specimen 1: NH0151948-0, *Cassipourea gummiflua* Tul. verticillata (N.E.Br.) J. Lewis (Figure 2).Specimen 3: NH0151950-0, *Cassipourea gummiflua* Tul. verticillata (N.E.Br.) J. Lewis.Specimen 4. NH0151951-0, *Cassipourea flanaganii* (Schinz) Alston.

The results obtained from the Liquid Chromatography–Mass Spectrometry (LC-MS/MS) yielded a total number of twenty-four compounds of different chemical classes, including fatty acids, steroids, di- and tri-terpenoids, flavonoids, phenolic acids were detected, and eighteen among them were tentatively identified (as summarized in Table 1).

The first eluted compound was identified as hexose/glucose with a mass of 215 *m*/*z* at 0.99 min which was already reported by (Sans et al., 2017) appeared in two plant extracts, along with lynoside (551 *m*/*z*), methyl linoleate (293 *m*/*z*), cassipourol (293 *m*/*z*), decahydroretinol (295 *m*/*z*) and emodin 6,8-dimethylether (297 m/z) which eluted at 4.83, 8.06, 9.53, 10.10 and 10.63 min, respectively. Lupeol eluted at 3.02 min and its derivative lupeol sterate (4.25 min) were identified with a mass of 425 *m*/*z* and 691 *m*/*z*, respectively [22,23,25]. Flavonoids such as isorhamnetin-3-*O*-rhamnoside (461 *m*/*z*), 2α,3α-Epoxyflavan-5,7,4′-triol-(4β → 8)-afzelechin (543 *m*/*z*), and tricin (329 *m*/*z*) eluted at 3.21, 5.53 and 6.98, respectively. Some of the compounds were terpenoids such as *ent*-atis-16-en-19-oic acid and *ent*-kaur-16-en-19-oic acid eluted at 8.49 and 12.35 min with mass 337 *m/z* and 309, respectively. According to the literature, azelaic acid [39,40] and tricin [41,42] were reported to inhibit tyrosinase significantly.

The identified compounds of the three methanolic extracts were based on their structure and molecular mass with a degree of similarity. Also, it has been predicted based on the compound structure reported in the previous reports with characteristic fragmentation patterns using a mass bank and a SciFinder database (Figure 3).

While the traditional names of the species are used interchangeably, a phytochemical comparison revealed that the three species are distinct, but they share skin-lightening properties.

*C. flanaganii* is a small scarce tree that occurs in forest patches between Qonce (King William’s Town) and southern KwaZulu-Natal in South Africa, used as a skin-lightening agent [1,43]. Its ground stem bark is mixed with water to form a paste and applied by black African females to their faces to enhance their beauty, it is known to clear blemishes, improve complexion, and lighten skin tone [12,19].

A recent study investigated the invivo toxicity of *C. flanaganii*. Both acute and sub-acute toxicity in Wistar rats were investigated. After the study period, after oral treatment with *C. flanaganii* crude stem bark extracts, acute or subacute toxicity symptoms were absent in Wistar rats at the levels administered. Liquid Chromatography–Mass Spectrometry (LC-MS) chemical profiling of the total extract identified eleven (11) compounds as the major chemical constituents [1]. It is well known that mercury-containing skin-lightening products can be absorbed through the skin and cause end-organ damage and that topical steroids can suppress the Hypothalamic-Pituitary-adrenal (HPA) axis after prolonged use [44,45]. As a result, it is critical to determine whether any topical treatment can cause systemic side effects by administering it orally and observing any systemic end-organ uptake. The oral exposure of laboratory animals to high doses of the test plant extract aids in determining potential hazards to humans who are accidentally exposed to much higher doses. In this study, it was determined that *C. flanaganii* extracts were non-toxic.

*Cassipourea malosana* is an evergreen tall tree distributed throughout African countries and used as a skin-lightening agent [27]. *C*. *malosana* is also reported to be closely related to, and often confused with, *C. flanaganii* Schinz (Alston) (Rhizophoraceae). Recently, *C. malosana* crude stem bark collected from Kenya was studied for its effects on tyrosinase. Eleven isolated compounds from the crude stem bark of *C. malosana* were studied for their cytotoxicity against a human ovarian cell line. Most of the test compounds showed no or weak cytotoxic activity. The isolated compounds showed little cytotoxicity against human ovarian cell line TOV21G, but the methyl derivatives of favan dimers exhibited higher activity than the parent compounds. Results from this study suggested that *C. malosana* bark is a potentially promising natural resource in the search for new bioactive agents [27].

*Cassipourea gummiflua* Tul. Verticillata is a small to large-size tree with dense foliage, dark brown to grey bark, and greenish-cream-coloured flowers, which grows up to 25 m tall [46]. It is a close relative of *C*. *gerradi* and is mostly found in the coastal forests of Northern Zululand [36]. Rondo [47] reported that the stem bark of the plant is said to be used as an alternative to *C*. *malosana* as a skin lightener and to treat skin ailments and sunburn. Also used for protection from evil spirits. No conclusive studies have been carried out on the chemical substances present in this plant species. However, a few phenolic compounds [36,48,49], organosulphur compounds [50] and alkaloids [46,51] have been identified as leaf constituents.

The harvesting and trade of plant material from rural communities for medicinal purposes has been and continues to be a contentious issue, especially in terms of biodiversity conservation [14]. Over-exploitation of plants for medicinal purposes for commercial trade endangers the survival of many species, as it is a stem bark that is harvested destructively, this results in death through ring barking of individual trees (Figure 4). Hence, conservation regulations and programs must be implemented. Increased public awareness would aid in the abolition of prejudices against medicinal plant production [1,12,14].

Hence, an understanding of their conservation status is important for guiding conservation policy development and action, contextualizing community-based natural resource management, and rural livelihood strategies. Short-term socio-economic gains are frequently prioritized over the long-term sustainability of both resources and traditional medicinal practices [15,18,20].

There is a growing demand and use of herbal cosmetics for skin purposes due to their perceived safety, formulation stability, efficacy, and rapid metabolism when applied to the skin [18]. According to ethnobotanical literature, topical application is the most commonly used mode of application because it ensures direct and immediate contact of the specific botanical compounds with the site of action [12,13,14,18]. Despite the recent popularity of modern cosmetic products, it is clear that plants continue to play an important role in the local cosmetics industry in South Africa’s Eastern Cape and KwaZulu-Natal community provinces. As a result, encouraging their sustainable use is a means of harnessing the conservation of these plants while also contributing to the local economy.

## 3. Materials and Methods

### 3.1. Identification of Plant Material

Using purposive sampling, knowledge holders were identified from the bus rank markets in Qonce (formerly known as King Williams Town) a town which is under Buffalo City Metropolitan Municipality and Bizana, a small town which is under the Winnie Madikizela Mandela Local Municipality, Alfred Nzo District Municipality, both towns are in South Africa Informants were knowledge holders and informal traders, who harvested the crude stems of the plants and sold them in the nearest marketplace, usually at a bus rank (Figure 5). The informants accompanied the researchers to the natural forests to identify various plant species that were used for complexion enhancement.

The crude stem is sold in two ways, as grounded/powdered or as crude bark. The plants were initially identified using their common names. Informal traders were predominantly elderly black men and women, including traditional healers.

### 3.2. Herbarium Specimen Preparation

Identified plant specimens were collected and mounted on herbarium sheets using glue and masking tape. Collected plant specimens were validated by the ethnobotanist, Professor Neil Crouch, from the South African National Biodiversity Institute (SANBI) and later deposited into the Botanic Gardens Herbarium, Durban, and received voucher specimens. The first three specimens were collected from the KwaMadiba location, Bizana, while the 4th was collected from Pirie Forests in Qonce (King Williams Town).

### 3.3. Preparation of the Plant Extracts

The crude stem bark was air dried under the shade, soaked into methanol, prepared into a paste, and then dried to powder as previously described by Mpofana et al., 2023 [1].

### 3.4. LCMS Equipment and Chemical Reagents

To estimate the antityrosinase efficacy of the plant material. Liquid chromatography mass spectrometry (LC-MS) Analysis. A Waters Synapt G2 Quadrupole time-of-flight (QTOF) mass spectrometer (MS) connected to a Waters Acquity ultra-performance liquid chromatograph (UPLC) (Waters, Milford, MA, USA) was used for the Liquid Chromatography–Mass Spectrometry (LC-MS) analysis. Electrospray ionization was used in negative mode with a cone voltage of 15V, desolvation temperature of 275 °C, desolvation gas at 650 L/h, and the rest of the MS settings optimized for best resolution and sensitivity. The specific negative ionization modes (*m*/*z* [M-H]^−^ or [M+Cl]^−^) were used to analyse the compounds.

## 4. Conclusions

Three plant species belonging to the Rhizophoraceae families were identified and documented as being used for skin lightening and UV protection cosmetic purposes among the Xhosa and Zulu women in the Eastern Cape from Bizana as well as Qonce (King Williams Town). The LC–MS/MS analysis showed that the three studied *Cassipourea* extracts were found to contain compounds that have anti-tyrosinase activity and consequently, it can be said that the anti-tyrosinase effect is due to the presence of these compounds, either as synergy or as individuals.

Medicinal plants continue to play an important role in the local cosmetics industry; therefore, their long-term use as well as large-scale cultivation as part of a formal Biodiversity Management Plan for this species should be encouraged. Given South Africa’s high unemployment and widespread poverty, we believe that indigenous communities should be assisted in the commercialization and related job creation associated with the economic development of the South African flora.

## Figures and Tables

**Figure 1 ijms-25-00237-f001:**
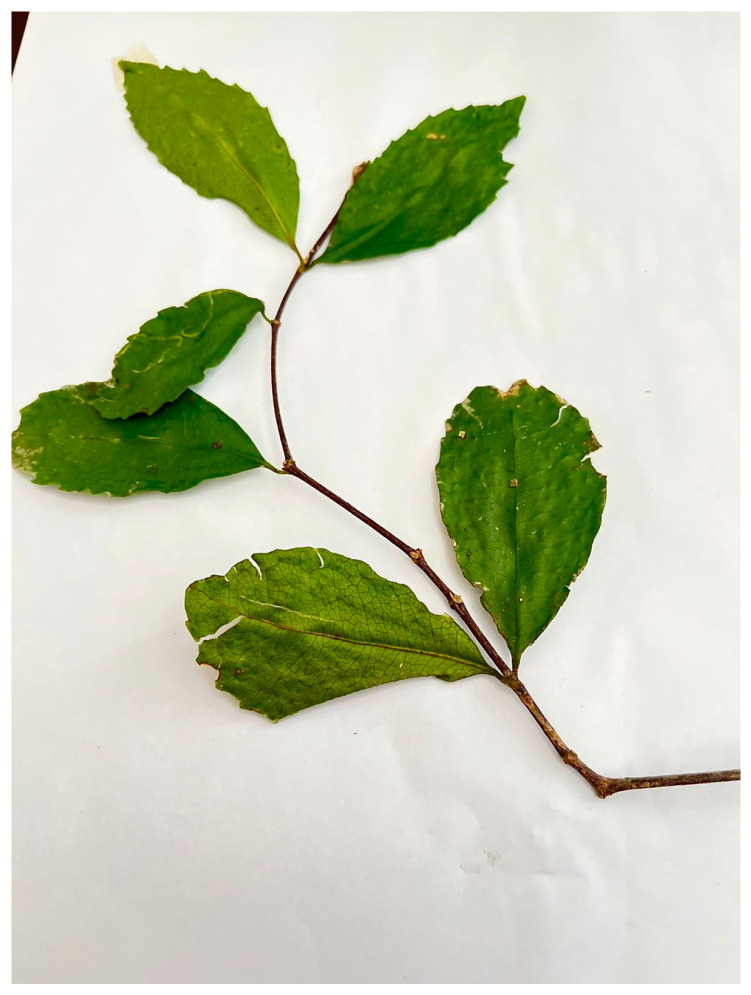
*Cassipourea malosana* (original image supplied by Nomakhosi Mpofana.

**Figure 2 ijms-25-00237-f002:**
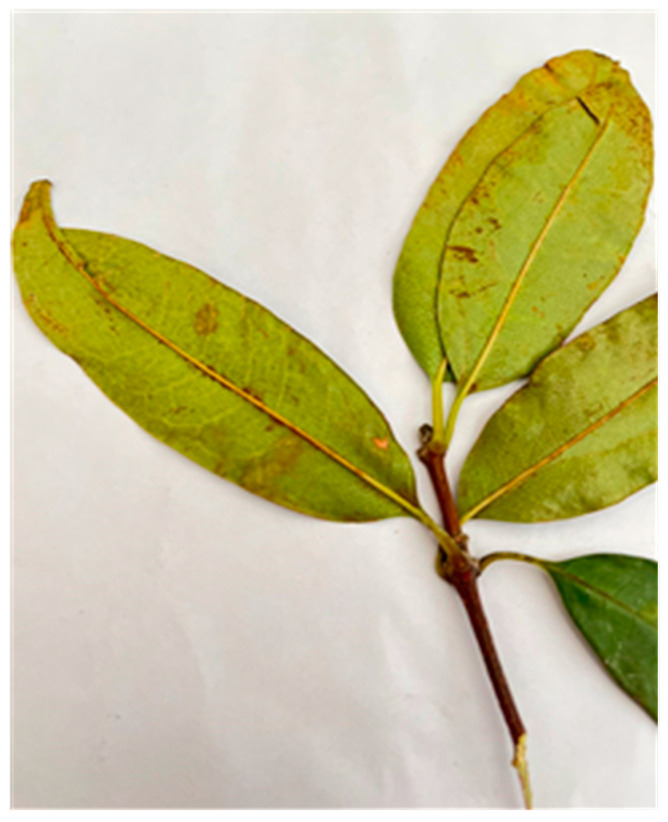
*Cassipourea gummiflua* var. *verticillata* (original image supplied by Nomakhosi Mpofana).

**Figure 3 ijms-25-00237-f003:**
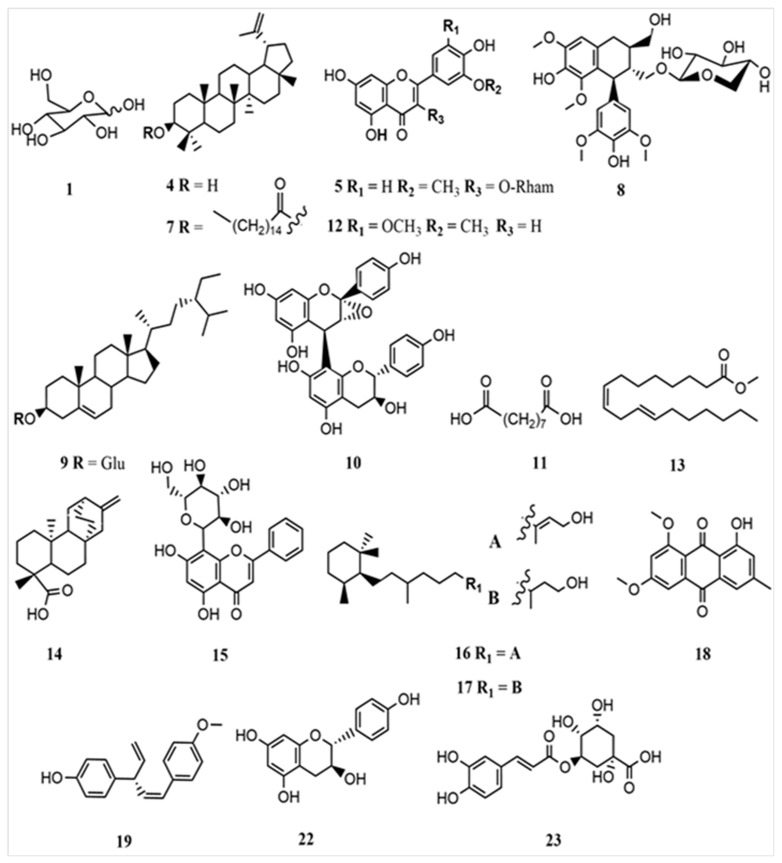
Compound elucidated from the three *Cassipourea* species.

**Figure 4 ijms-25-00237-f004:**
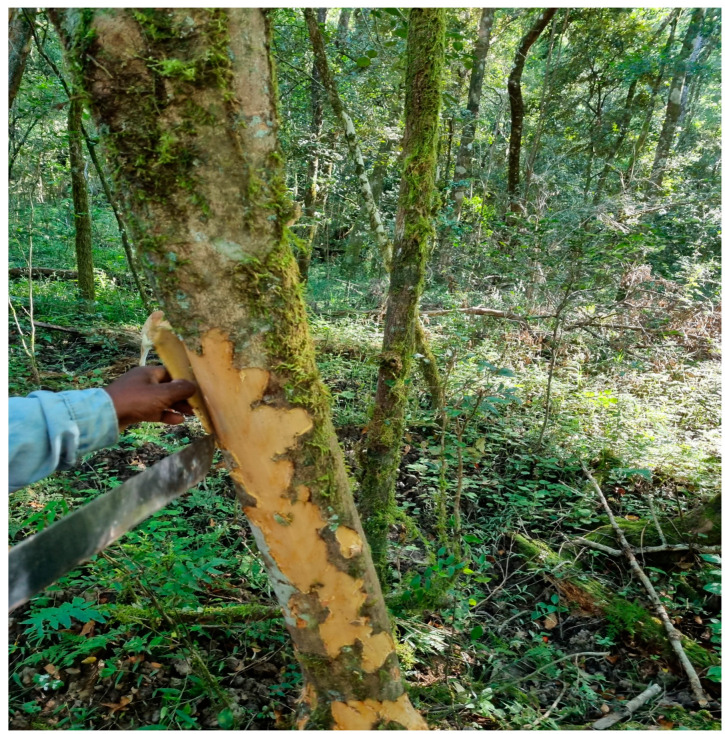
Illustration of the stem bark harvesting demonstrating the potential to destroy the plant (original image supplied by Nomakhosi Mpofana).

**Figure 5 ijms-25-00237-f005:**
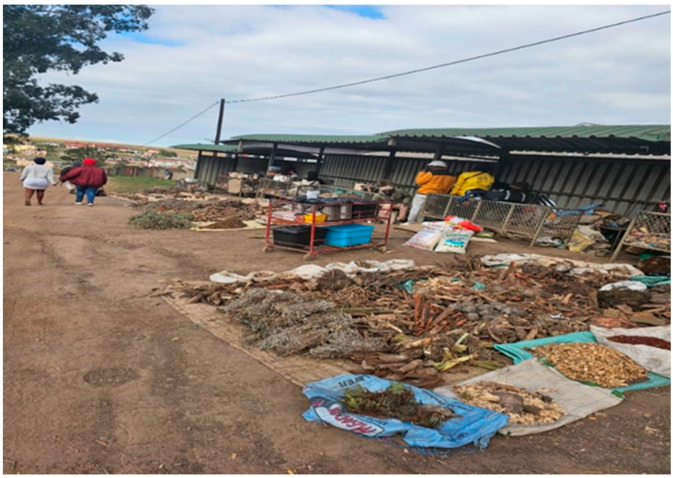
A marketplace in Bizana at the bus rank where informal traders sell various plant parts for different ailments and disorders (original image supplied by Nomakhosi Mpofana).

**Table 1 ijms-25-00237-t001:** Phytochemical comparison of three *Cassipourea* species, *C*. *flanaganii*, *C*. *gummiflua* and *C*. *malosana* by LC-MS/MS analysis in negative mode.

Peak	Proposed Compound	*m*/*z*	t_R_ (min)	[M-H]^−^	Molecular Formula	*C*. *flanaganii*	*C*. *gummiflua*	*C*. *malosana*	Refs.
1	Hexose (glucose)	215.033	0.99	[M+Cl]^−^	C_6_H_12_O	+	-	+	[21]
2	Unknown	194.1	1.15	[M-H]^−^	Unknown	-	+	-	-
3	Unknown	341.1	1.21	[M-H]^−^	Unknown	-	+	-	-
4	Lupeol	425.075	3.02	[M-H]^−^	C_30_H_50_O	+	+	+	[22,23]
5	Isorhamnetin-3-O-rhamnoside	461.129	3.21	[M-H]^−^	C_22_H_22_O_11_	+	-	-	[24]
6	Unknown	252.014	3.55	[M-H]^−^	Unknown	+	-	+	-
7	Lupeol stearate	691.202	4.25	[M-H]^−^	C_48_H_84_O_2_	-	-	+	[25]
8	Lyoniside	551.202	4.83	[M-H]^−^	C_27_H_36_O_12_	+	+	-	[12]
9	Sitosterol glycoside	575.103	5.53	[M-H]^−^	C_35_H_60_O_6_	-	-	+	[26]
10	2α,3α-Epoxyflavan-5,7,4′-triol-(4β → 8)-afzelechin	543.127	5.55	[M-H]^−^	C_30_H_24_O_10_	-	+	-	[27]
11	Azelaic acid	187.097	5.62	[M-H]^−^	C_9_H_16_O_4_	+	+	+	[28]
12	Tricin	329.232	6.98	[M-H]^−^	C_17_H_14_O_7_	+	+	+	[29]
13	Methyl linoleate	293.138	8.06	[M-H]^−^	C_19_H_34_O_2_	+	-	+	[30]
14	ent-atis-16-en-19-oic acid	337.1	8.49	[M+Cl]^−^	C_30_H_24_O_10_	-	+	-	[12]
15	Chrysin 8-C-glucoside	415.103	9.09	[M-H]^−^	C_27_H_44_O_3_	-	+	-	[31]
16	Cassipourol	293.211	9.53	[M-H]^−^	C_20_H_38_O	+	-	+	[32]
17	Decahydroretinol	295.228	10.10	[M-H]^−^	C_20_H_40_O	+	-	+	[32]
18	Emodin 6,8-dimethyl ether	297.242	10.63	[M-H]^−^	C_17_H_14_O_5_	+	-	+	[33]
19	Ellisinin A	265.1	11.32	[M-H]^−^	C_18_H_18_O_2_	-	+	-	[34]
20	Unknown	311.2	11.48	[M-H]^−^	Unknown	-	+	-	-
21	Unknown	327.256	11.63	[M-H]^−^	Unknown	+	-	-	-
22	Afzelechin	309.2	12.35	[M+Cl]^−^	C_15_H_14_O_5_	-	+	-	[35,36]
23	Chlorogenic acid	353.2	12.76	[M-H]^−^	C_16_H_18_O_9_	-	+	-	[37,38]
24	Unknown	397.2	12.91	[M-H]^−^	Unknown	-	+	-	-

## Data Availability

All data is included in the manuscript.

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
