# Peer review of "Analysis of Three Species of Cassipourea Traditionally Used for Hypermelanosis in Selected Provinces in South Africa"

_ijms, 2023, doi:10.3390/ijms25010237_

Round 1

Reviewer 1 Report

Comments and Suggestions for Authors

 Analysis of Plants Traditionally Used for Hypermelanosis in South Africa

Hussain et al.

Authors must review certain aspects in the work

Page 3

Say: Sci-finder          Should say:  SciFinder

Page 4,

Table 1. Please write plant names in italic style in the manuscript and the bibliography: Cassipourea species, C. flanaganii, C. gummiflua and C. malosana

Peak 8. Did you mean Lyoniside?

Peak 13- Did you mean Linoleate?

Peak 15- It is true that the plant has cholestenoic acid?

In Table 1, the name of 17 compounds is indicated, not of 18.

Line 95, mahaunnin B. Did you mean Mahuannin B ?. I do not find this compound in table 1.

Please revise the bibliography, commas/semicolons after the name in ref 27, and the names in italics in some plants. Shouldn't they put the names of the magazines in the abbreviated form?

Author Response

Please find attached the point by point response to the reviewer 1. 

Thank you.

Reviewer 2 Report

Comments and Suggestions for Authors

Thank you for the opportunity to read and prepare a review of your manuscript entitled: Analysis of Plants Traditionally Used for Hypermelanosis in South Africa. It is a paper of a brief communication nature on an interesting topic. The Authors identified the chemical composition of three plant species traditionally used in skin lightening and photoprotection. The authors' conclusions about the possibility of activating local people to obtain raw materials what can be an economically activating element seems doubly interesting.

However, the text needs numerous corrections.

1. please prepare an abstract that is less about the conclusion of economic and social significance and more related to the experiment conducted. Please remember to explain every abbreviation used for the first time. This also applies to the abstract.

2. please check the entire text very carefully and improve the spelling of Latin names - they should be written in italics. The first time a species is mentioned in the text, it must have a full botanic (3 element) name. Each subsequent time: abbreviated name (first letter from the group name + species epithet).

3. please remove all personal forms of speech

4 Please change the layout of the paper. Literature sources should not be indicated in the section: results. Therefore, I propose that the authors change the section: results to Results and Discussion. This combination will allow the presentation of results and discussion.

5 Figure 1 is completely redundant. It is also an external material. The authors must have written permission from the publisher to republish them. Therefore, I propose to remove this graphic form and describe the issue.

6. please do not indicate that plant raw materials are completely free of adverse reactions. This is erroneous thinking and cannot be part of a scientific publication. Please remember that nicotine, cocaine and numerous poisons and toxins are components of plant origin. Any ingredient can give side effects especially at a time when we are seeing a huge increase in people with atopia.

7 Please change the title of your paper. The current one indicates that the manuscript will deal with a number of species, while the paper deals with only 3 species in addition to related ones. I also ask that the geographical area be narrowed down - the authors themselves indicated in the text that they dealt with garuna used in a specific geographical area of Africa. Please also correct the term "country" when referring to Africa.

8. figure 4 needs to be improved - please prepare the patterns in one graphics program. In its current form it is unsightly.

9. bibliographic item No. 4 has been incorrectly transcribed

to sum up: the paper requires writing an abstract de novo, changing the layout of the work, and corrections regarding the correct form of the notation of the species names of the plants discussed.

Author Response

Attached please find the point by point response to the reviewer 2 comments.

Thank you

Round 2

Reviewer 2 Report

Comments and Suggestions for Authors

Thank you for making the corrections. Would the Authors please consider a suggestion that the modified title read as follows: "Analysis of three species of Cassipourea traditionally used for Hypermelanosis in selected provinces in South Africa". 

Author Response

Please find attached the responses to reviewer comments.

Thank you
